# Development of a Tool to Assess the Implementation and Perception of the Value-Based Healthcare Model: Our Results

**DOI:** 10.3390/healthcare13222849

**Published:** 2025-11-10

**Authors:** Egidio de Mattia, Dorine van Staalduinen, Ilaria Valentini, Paul B van der Nat, Antonio Giulio de Belvis

**Affiliations:** 1Faculty of Economics, Università Cattolica del Sacro Cuore, 00168 Rome, Italy; ilaria.valentini@unicatt.it (I.V.); antonio.debelvis@policlinicogemelli.it (A.G.d.B.); 2Critical Pathways and Evaluation Outcome Unit, Fondazione Policlinico Universitario “A. Gemelli”—IRCCS, 00168 Rome, Italy; 3Santeon, 3584 AA Utrecht, The Netherlands; d.van.staalduinen@antoniusziekenhuis.nl (D.v.S.); p.van.der.nat@antoniusziekenhuis.nl (P.B.v.d.N.); 4Martini Hospital, 9728 NT Groningen, The Netherlands; 5Department of Value-Based Healthcare, St. Antonius Hospital, 3435 CM Nieuwegein, The Netherlands; 6Scientific Center for Quality of Healthcare (IQ Health), Geert Grooteplein Zuid 10, 6525 GA Nijmegen, The Netherlands

**Keywords:** value-based healthcare, tool, change management, leadership

## Abstract

Background/Objectives: A successful implementation of VBHC requires significant transformations, spanning from structural adjustments to a fundamental shift in organizational culture. However, organizations often adopt VBHC principles piecemeal, while managers overlook diverse staff reactions and their leadership role. To address these challenges, we developed and validated a multi-source assessment tool for use by organizations during their transition to VBHC to determine whether they are progressing effectively in both the dimensions. Method: To develop the first version of the tool, a multidisciplinary group translated strategies identified in a previous scoping review into questionnaire items through structured sessions. To validate it, a two round Delphi analysis was conducted. Experts in VBHC, management, health economy and clinical/researcher were contacted to ask for their participation. Consensus was established based on the following criteria: a median (Mdn) score of ≥4, an interquartile range (IQR) of ≤1.5 or ≤2, and a frequency of ratings in the range [4–5] ≥ 70%. Results: A total of 54 experts were invited to participate in the Delphi survey. The first round received 25 responses, while the second round had 23 responses. The final version of the assessment tool consists of 30 questions divided into two macro-areas: the Value Agenda Section and the Change Management Section. All items achieved a minimum frequency of ratings > 80% during both the rounds. Conclusion: By comparing the perspectives of managers and clinicians, the resulting tool enables organizations to assess the adoption of the Value Agenda component, as well as the change management strategies supporting its implementation.

## 1. Introduction

In recent decades, healthcare systems worldwide have faced growing pressure to enhance efficiency, improve patient outcomes, and ensure financial sustainability. Rising healthcare expenditures, variations in care quality, medical errors, and persistent barriers to access underscore the urgency need for transformation.

In response, Value-Based Healthcare (VBHC) has emerged as a paradigm shift that reorients care delivery from a volume to value [1,2,3]. Introduced by M.E. Porter and E.O. Teisberg in 2006 [2], VBHC defines value as the “health outcomes that matter to patients divided by the costs needed to achieve them” [2]. This framework challenges traditional, specialty-based organization models and promotes patient-centered systems that emphasize integration, multidisciplinary collaboration, and continuity of care [4,5,6,7]. Despite its conceptual appeal, VBHC implementation remain complex, requiring both structural and culture transformation [8,9].

To guide this transformation M.E. Porter and T.H. Lee introduced the “Value Agenda” [10], which comprises six interdependent elements. Its cornerstone is the reorganization of care around patients’ medical conditions through Integrated Practice Units (IPUs). The agenda also highlights systematic measurement of outcomes and costs, the development of care networks to deliver the right services in the right settings, and the use of integrated Information technology (IT) platforms to coordinate all components [2]. In terms of reimbursement, the agenda advocates shifting from the DRG payment system to bundled payments for care cycles, aligned with value creation [2].

Although many healthcare providers worldwide have begun adopting VBHC principles [11,12,13], several studies indicate that few have implemented VBHC as an integrated management strategy [13,14]. Most have selectively focused on elements that fit with existing organizational structure. This selective approach inhibits the development of a truly value-oriented care model, particularly when considering the van der Nat’s extended strategic agenda [15]. According to him, for VBHC to become fully operational, the Value Agenda needs to be extended with four additional elements: setting up value-based quality improvement processes, integrating value into patient communication, investing in a culture focused on value delivery, and creating learning platforms for healthcare professionals [15].

The implementation of these elements profoundly affects established practices and processes. Consequently, if such structural and transformative interventions are not accompanied by a strategy on cultural change, their successful implementation will be compromised [9]. Organizational culture, shaped by shared values and behavioral norms, often represents the greatest barrier to VBHC adoption [16]. Therefore, leaders must actively promote a VBHC-oriented culture through targeted change management strategies that engage staff and promote understanding of the benefits of transformation [17]. Coetsee et al. [17] identified five strategies to achieve what Weiner et al. [18] defined as “organizational readiness” (OR), namely “the degree to which organizational members are psychologically and behaviorally prepared to implement change”. These strategies include fostering a learning environment, ensuring transparent communication, empowering staff participation, recognizing and rewarding performance, and promoting a shared vision that aligns individual and organizational goals. Similarly, J. P. Kotter [19] outlined eight essential steps for successful organizational transformations, from creating a sense of urgency and forming a powerful coalition to anchoring new approaches in organizational culture.

However, many managers tend to underestimate the diversity of employee responses to organizational change and their role of leadership in shaping them [16]. This lacuna, combined with the “piece-meal adoption” of the VBHC tenets [20], remains a major barrier to comprehensive implementation.

The present study aims to develop and validate an assessment tool that organizations can use during their transition to VBHC with a dual aim:(1)To assess the extent to which the Value Agenda elements have been implemented within an organization, and(2)To evaluate the change management strategies employed by organizational leaders to drive and sustain the transition to VBHC.

## 2. Materials and Methods

The study was conducted between 2023 and early 2025 and consisted of 2 phases. a preparatory literature search and experts’ involvement (phase 1: August 2023–November 2024) followed by a survey using the Delphi method (phase 2: November 2024–January 2025). This article presents phase 2, while the methodology and findings from of phase 1 were published elsewhere [9].

### 2.1. Phase 1: Tool Development

The present assessment tool builds upon the findings of our previous scoping review [9], which aimed to identify the operational and management strategies employed at various levels—healthcare policymakers, hospital management, and healthcare providers—to implement VBHC principles in real-world settings. For this assessment tool, the focus is narrowed to strategies within hospital boundaries, specifically addressing change management and implementing the Value Agenda.

To translate operational strategies into practical questionnaire items, a series of structured sessions were organized between September 2024 and November 2024. This process involved a multidisciplinary working group composed of 12 experts in VBHC, change management, hospital operations, clinical care, and statistics.

### 2.2. Phase 2: Tool Validation

To validate the initial version of the assessment tool developed in Phase 1, a two-round Delphi survey was conducted. The Delphi survey involved the following steps: (i) Development of an online survey; (ii) Recruitment and consenting of participants to the Delphi panel; (iii) Two rounds of consultation on the proposed topics in the survey.

#### 2.2.1. The Panel Members

For the Delphi panel, potential members, who spoke English, were selected among international experts based on their Curriculum Vitae, scientific publications, and demonstrated expertise in the following domains: (i) VBHC, (ii) healthcare management, (iii) health economics, and (iv) clinical practice and research. Experts were identified through literature searches, professional networks, and members of learning communities.

Experts were invited to complete the Delphi survey via email, using a Google Forms questionnaire. A cover letter outlined the survey’s purpose, relevance, and significance. Responses were collected in real-time and anonymously. Two reminder emails were sent at one-week intervals to increase the response rate. Expecting a response rate in the range of 30–40 percent, a total of 54 experts were invited.

#### 2.2.2. Data Analysis and Definition of Consensus

The experts panel assessed the adequacy of the proposed items using a 5-point scale, ranging from 1 (not adequate) to 5 (completely adequate). Data collected during the initial consultation round were used to calculate the Kendall’s W coefficient and establish consensus criteria among panelists. Consensus was establishing based on the following criteria: a median (Mdn) score of ≥4; an interquartile range (IQR) was used as a non-parametric consensus metric; lower IQR values indicate tighter agreement among experts. A priori, consensus was defined as IQR ≤ 1.5 (primary threshold), with IQR ≤ 2 adopted as a lenient criterion in sensitivity checks; and a frequency of ratings in the range [4–5] ≥ 70% [21].

Items meeting inclusion criteria in the first round were carried forward to the second round, where their stability within expert consensus was re-evaluated. The purpose of round 2 was to confirm the robustness of the agreement obtained in round 1. The final set of items included those that met the consensus criteria in both rounds.

To further validate the findings, several complementary statistical analyses were performed:Central tendency and dispersion analysis;Consensus evaluation between rounds: To gauge agreement, we relied on the IQR, a non-parametric measure in which smaller values reflect tighter convergence of ratings and thus greater consensus. We complemented this with the Relative Interquartile Range (RIR), which expresses how dispersed the responses are in relation to the central tendency; it is computed as (IQR/Median) × 100, so lower RIR values indicate that judgments cluster more closely around the median. To assess how stable the panel’s views remained across rounds, we examined the Variation in the Coefficient of Variation (VCV), defined as the percentage change in the coefficient of variation (CV) from Round 1 to Round 2. The CV is calculated as (Standard Deviation/Mean) × 100, and VCV as ((CV_Round2 − CV_Round1)/CV_Round1) × 100. In this framework, smaller—and especially negative—VCV values signal that variability decreased over time, consistent with increasing stability of expert judgments across successive Delphi rounds.Reliability assessment using Cronbach’s alpha;Stability analysis of group responses through the median test and U test.

This methodology replicated one already applied in an another study [21].

## 3. Results

### 3.1. Tool Development

The initial pool of items was derived from the operational strategies identified in the scoping review. After three rounds of expert review and refinement meetings, redundant or unclear items were removed, resulting in a final 30-item questionnaire structured into two main sections: the Value Agenda (13 items) and Change Management (17 items). Each section is subdivided into thematic sub-areas (4 and 5, respectively), with mirrored items designed for both healthcare managers and frontline professionals.

### 3.2. Tool Validation

A two-round modified Delphi study was carried out for this study.

Panel Participation

A total of 54 experts were invited to participate in the Delphi survey. The first round received 25 responses (46%), while the second round had 23 responses (43%) (Table 1). This slight decrease in participation is consistent with the Delphi methodology, where attrition between rounds is expected as participants refine their evaluations.

Value Agenda Section

In the first round (R1), most items in this section received ratings concentrated in the 4–5 range, indicating strong agreement among panel members. However, some priorities had a wider distribution of responses, prompting further refinement. The second round (R2) showed a shift toward greater consensus, with a reduction in response variability. Items that initially had more dispersed ratings (including scores of 2 and 3) saw an increase in higher agreement (4–5 scores) (Figure 1). These findings indicate that after initial feedback, experts refined their evaluations, leading to a stronger collective endorsement of the priorities. The level of agreement among the 13 items, measured using Kendall’s coefficient of concordance (W), increased slightly between Delphi rounds—from W = 0.196 (χ^2^(24) = 61.1, *p* < 0.001) in Round 1 to W = 0.222 (χ^2^(22) = 63.6, *p* < 0.001) in Round 2. The statistically significant results indicate a progressive trend toward consensus among panelists.

Change Management Section

Compared to the Value Agenda section, the Change Management section exhibited slightly more variability in responses during the R1 with some items receiving scores across the full 1–5 scale. This suggests that panel members had differing perspectives on the importance or feasibility of certain priorities within this domain. In the R2, consensus improved for several key priorities, with a notable decrease in lower ratings (1–2 scores) (Figure 2). This indicates that after reviewing collective input, panel members aligned more closely on the critical importance of change management strategies. However, a few priorities continued to display some variability, suggesting the need for further discussion or clarification in future iterations. In the Change Management domain, the level of agreement among the 17 items, assessed using Kendall’s coefficient of concordance, increased from W = 0.208 (χ^2^(24) = 84.7, *p* < 0.001) in Round 1 to W = 0.280 (χ^2^(22) = 105, *p* < 0.001) in Round 2. Although overall concordance remained moderate, the significant increase in W indicates a progressive alignment of expert opinions across rounds.

Data Analysis

The analysis of responses collected during the R1 revealed broad agreement among panelists regarding the two proposed dimensions. As a result, all items were included in the R2 for further assessment. After R2, the selection criteria for the final items were specified through an interpretation of the contingency table with consensus results. Highly significant items demonstrated unanimous consensus across both rounds, while non-significant items were excluded at each stage of evaluation.

The final set of questions (Figure 3) is presented in Table 2.

○Analysis of Consensus Level and Group Stability in Answers

To assess the proximity and stability of responses between rounds, the IQR and VCV were calculated. The analysis indicated an acceptable degree of proximity and stability among panellists’ opinions (IQR < 0.5).

Group stability was confirmed when the RIR between rounds was < 0.20. Additionally, consensus was established when the CV remained below 22% for most items.

Appendix A shows that the VRIR values for all items remained below the 0.30 threshold, confirming stability in responses. Consensus among panelists is further substantiated by the VCV data.

The Delphi process was deemed complete once consensus and stability levels were reached, as additional rounds would not have produced significant variations in the results.

○Questionnaire Validity and Reliability Analysis

The reliability of the questionnaire was assessed using Cronbach’s alpha, which yielded a high overall score of 0.962, confirming strong internal consistency. Individual reliability scores for each dimension exceeded 0.8, confirming internal consistency [R1: Value Agenda section = 0.822; Change Management section = 0.884; R2: Value Agenda section = 0.876; Change Management section = 0.932].

An item-total correlation analysis was performed, revealing an optimal inter-item correlation range. The removal of any individual items did not enhance reliability, indicating a robust instrument. Inspection of “alpha if item deleted” indicated no item would increase α; differences were trivial, confirming that item removal would not yield a meaningful gain in reliability. Accordingly, all items were retained.

○Independent Samples *t*-Test

To compare responses between rounds, an independent samples Mann–Whitney U test was performed. The results, presented in Appendix A, indicate that no statistically significant differences were found between the rounds (*p* > 0.05), further reinforcing the stability of panelist responses over time.

## 4. Discussion

In this paper, we presented the development and validation of an assessment tool designed to support healthcare organization in their transition towards VBHC. We called it “Embrace Value-Driven Change”. This tool serves a dual aim: (i) to assess the extent to which the Value Agenda elements have been implemented within an organization, and (ii) to evaluate the change management strategies employed by organizational leaders to drive and sustain this transition.

As structured, the tool incorporates innovative features.

The first distinctive feature lies in its design. The final version consists of 30 items divided into two sections: 13 items in the Value Agenda Section and 17 in the Change Management Section. Each area includes “main questions” (e.g., “To what extent does the hospital systematically measure clinical and patient reported outcomes?”), followed by “secondary questions” (e.g., “If so, which of the following measures are used?”), which explore specific aspects of each topic. To safeguard objectivity and reproducibility, multiple choice questions were chosen.

To capture potential misalignment between hospital top management and healthcare personnel, the tool was designed as mirror survey. For instance, hospital managers are asked: “To what extent are members of the organization regularly updated on the progress of the VBHC plan over time?”, while healthcare personnel are asked the mirror question: “To what extent are you informed about the progress of such VBHC plan over time?”. This design, which has been already employed in other settings [53,54,55,56], enables a comprehensive understanding of how VBHC is being implemented and helps identity whether further alignment ore corrective actions are needed.

The second feature lies in the scope of investigation. Existing instruments typically focus on either the implementation of value agenda elements or the organizational strategies for change management; none integrate both dimensions within a single framework. For example, the questionnaire developed by H.J. Westerink et al. [57] captures the perspective of care teams and focuses solely on the structural adoption of Value Agenda components. In contrast, our tool collects data from both managers and clinicians and also explores the change management strategies underpinning VBHC implementation.

Although the content of the first section, albeit expressed differently (e.g., ”To what extent are outcomes measures and casemix variables structurally being measured for the medical condition?” versus “To what extent does the hospital systematically measure clinical and patient reported outcomes?”), partly overlaps with Westerink’s instrument (for instance, in evaluating the organization of care around medical conditions or the measurement of outcomes and costs), our tool further examines whether these measures are shared internally to inform quality improvement processes and communicated to patients to promote engagement and shared decision-making. In line with van der Nat’s extended strategic agenda [15], these aspects reflect the integration of value into patient communication and the creation of a value-oriented culture. However, our tool does not yet include two key pillars of van der Nat’s extended strategic agenda: build learning platforms for healthcare professionals and aligning reimbursement with value. These elements may be included in future iterations of the tool.

On the change management side, numerous frameworks have been developed to help managers evaluate their change initiatives. One example is the DICE framework. This framework, developed by H.L. Sirkin et al. [58], evaluates change initiatives based on duration, integrity, commitment, and effort. Unlike such models, our tool measures, from both top management and healthcare personnel perspectives, the implementation of key actions and strategies identified in our previous scoping review [9] as critical for successful change management. These strategies are consistent with those proposed by Coetsee [17] and Kotter [19], spanning from creating a sense of urgency (“To what extent did your organization establish a sense of urgency to implement the principles of VBHC?”), forming a guiding coalition (“To what extent was a formal group of people—key leaders and representatives from clinical and managerial teams—brought together to drive the implementation of VBHC?”) to developing and communicating a vision for the project (“a future-oriented perspective that emphasizes the reasons why you should strive to create that future”). The tool also investigates how organizations anchor VBHC principles in their culture through education, training and career progression opportunities. Finally, it explores barriers to implementation, asking respondents to identify the main obstacles encountered during the transition to VBHC (“Which were the main obstacles you encountered during the implementation of VBHC?”).

### Limitations

The development of this assessment tool could have been affected by some limitations. The first is related to the methodology chosen to validate it. We used the Delphi methodology. It ensured methodological rigor and expert consensus on the instrument’s key elements [59], but it has its own inherent limitations. Although this provides an evidence-based basis, we may have omitted elements that were not included in the scoping review but were worthy of inclusion in our tool, such as the role of the hospital in contracting for value-based payment mechanisms and the presence of platform-based learning systems. Future versions could explore how these missing pillars might be integrated or further examined to expand the instrument’s scope.

Second, the Delphi methodology is susceptible to high dropout rates due to the length of commitment and distractions between rounds. In our case, the first round received 25 responses, while the second round had 23 responses.

Finally, although the panel of experts included were of international standing, the majority of respondents (72%) were based in Italy. This limited geographical representation may have influenced the results. Future validations should aim for a more balanced international composition and include a greater proportion of non-clinical stakeholders such as managers, health economists, and representatives of patient organizations.

## 5. Conclusions

The resulting evidence-based assessment tool was designed to enable organizations to monitor the progress of VBHC adoption and evaluate the effectiveness of the change management strategies employed.

Its dual-perspective structure facilitates the assessment of the alignment between leadership and clinical teams, ensuring that the transition is both strategically driven and operationally embraced. This approach empowers managers to proactively steer change, address resistance, and enhance organizational readiness, ultimately fostering more effective and sustainable VBHC adoption.

## Figures and Tables

**Figure 1 healthcare-13-02849-f001:**
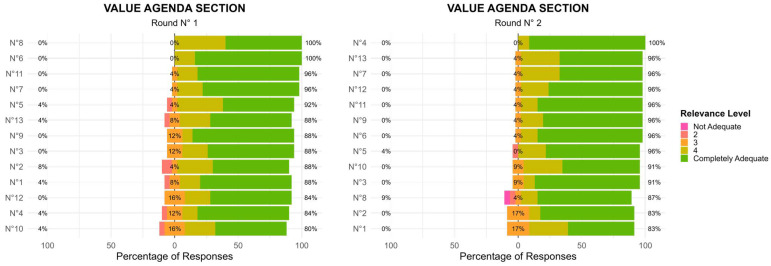
Value Agenda Section: Two-Round Delphi Results.

**Figure 2 healthcare-13-02849-f002:**
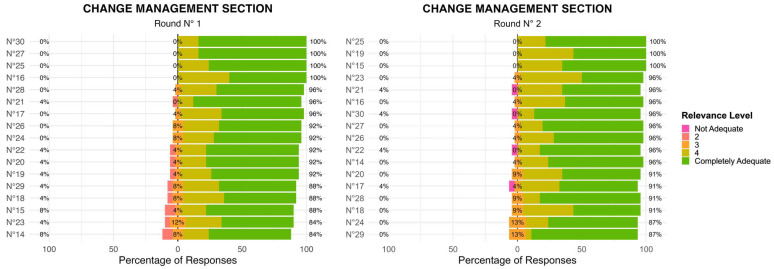
Change Management Section: Two-Round Delphi Results.

**Figure 3 healthcare-13-02849-f003:**
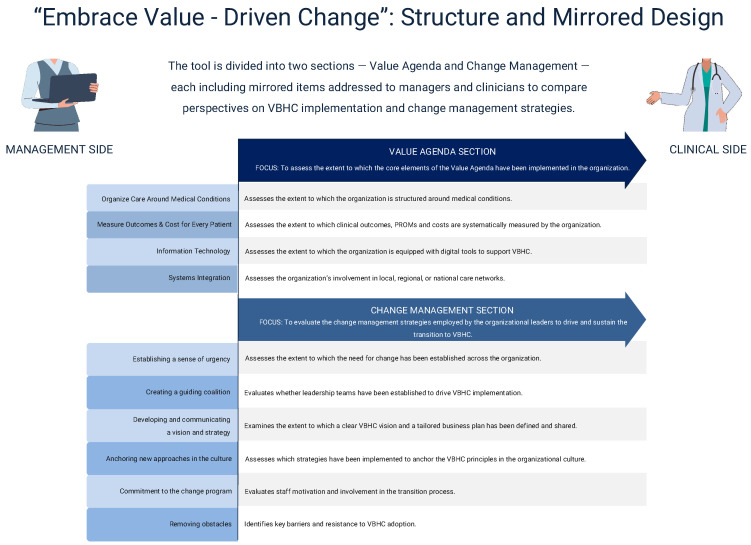
”Embrace Value-Driven Change” assessment tool.

**Table 1 healthcare-13-02849-t001:** Panel composition and expertise.

	Round 1		Round 2	
Gender	n	%	n	%
Female	14	56%	12	52%
Male	11	44%	11	48%
Country or region	n	%	n	%
Italy	18	72%	17	74%
The Netherlands	2	8%	2	9%
Brazil	2	8%	1	4%
Belgium	1	4%	1	4%
UK	1	4%	1	4%
Saudi Alaskar	1	4%	1	4%
Job title	n	%	n	%
Researcher	8	32%	6	26%
Medical Doctor	9	36%	9	39%
Manager	5	20%	5	22%
Professor	3	12%	3	13%

**Table 2 healthcare-13-02849-t002:** “Embrace Value-Driven Change” assessment tool.

Sub Area	Items	Prospective	Prospective	Answers	Sources
Value agenda section Evidence highlights that many organizations do not adopt the VBHC model as an integrated strategy; instead, they implement only selected elements that best align with their organizational structure. This partial approach hinders the model’s effectiveness, preventing the creation of a care system that maximizes value for the patient. Therefore, this section aims to explore which structural VBHC elements have been adopted by the organization and to what extent.					
Organize Care Around Medical Conditions	1	To what extent is care delivery in your hospital organized around medical conditions?	To what extent is care delivery in your hospital organized around medical conditions?	1 = Not at all 2 3 4 5 6 7 = To a full extent	[10,15,22,23,24,25,26,27,28,29]
	2	To what extent does the hospital structurally involve patients and/or caregivers in the condition-based teams?	To what extent does the hospital structurally involve patients and/or caregivers in the condition-based teams?	1 = Not at all 2 3 4 5 6 7 = To a full extent	[15,30]
	3	To what extent is the hospital equipped with support staff to assist condition-based teams in delivering care and continuously improving their quality of care delivery? (e.g., Data Analysts, Project Leaders, Care/case Managers, Clinical Coordinators, Quality Improvement Specialists, Patient Navigators)	To what extent is the hospital equipped with support staff to assist you and your team in delivering care and continuously improving quality of care delivery? (e.g., Data Analysts, Project Leaders, Care/case Managers, Clinical Coordinators, Quality Improvement Specialists, Patient Navigators)	1 = Not at all 2 3 4 5 6 7 = To a full extent	[15,24,29,31,32,33]
Measure Outcomes & Cost for Every Patient	4	To what extent does the hospital systematically measure clinical and patient reported outcomes?	To what extent does the hospital systematically measure clinical and patient reported outcomes?	1 = Not at all 2 3 4 5 6 7 = To a full extent	[10,15,23,24,25,26,31,32,33,34,35,36,37,38,39,40,41]
	5	If so, which of the following measures are used? (Select all that apply)	If so, which of the following measures are used? (Select all that apply)	Volume indicators Process indicators Outcomes indicators PREMs PROMs Other (specify) None of the above	[10,15]
	6	If so, to what extent does this clinical performance measurement system integrated in mechanisms for Audit & Feedback?	If so, To what extent does this clinical performance measurement system integrated in mechanisms for Audit & Feedback?	1 = Not at all 2 3 4 5 6 7 = To a full extent	[15,25,37]
	7	If so, at the end each audit meeting, to what extent are these improvement actions defined and shared, along with their respective responsible actors/figures and implementation timelines?	If so, at the end of each audit meeting, to what extent are these improvement actions defined and shared, along with their respective responsible actors/figures and implementation timelines?	1 = Not at all 2 3 4 5 6 7 = To a full extent	[25,37]
	8	To what extent are the results of these analyses (PREMS, PROMs and clinical outcomes) shared with the patients and/or caregivers?	To what extent are the results of these analyses (PREMS, PROMs and clinical outcomes) shared with the patients and/or caregivers?	1 = Not at all 2 3 4 5 6 7 = To a full extent	[15]
	9	To what extent does the hospital have a planning and control system capable of monitoring the costs of care delivery in condition-based teams?	To what extent does the hospital have a planning and control system capable of monitoring the costs of care delivery in condition-based teams?	1 = Not at all 2 3 4 5 6 7 = To a full extent	[42]
	10	If so, what type of cost accounting is used? (Select all that apply)	If so, what type of cost accounting is used? (Select all that apply)	Traditional Cost accounting method Activity-Based Costing (ABC) Time-Driven Activity-Based Costing (TDABC) Unsure Other (please specify):	[42]
Information Technology	11	To what extent does the hospital have an Electronic Medical Record (EMR) system?	To what extent does the hospital have an Electronic Medical Record (EMR) system?	1 = Not at all 2 3 4 5 6 7 = To a full extent	[8,10,23,31,32,33,35,43,44,45,46]
	12	To what extent is the hospital equipped with digital platforms (dashboard/app, etc.) that support clinicians by collecting outcome data in real time?	To what extent is the hospital equipped with digital platforms (dashboard/app, etc.) that support clinicians by collecting outcomes outcome data in real time?	1 = Not at all 2 3 4 5 6 7 = To a full extent	[30,45,47]
Systems Integration	13	To what extent is the hospital actively involved in local, regional or national networks aimed at developing condition-based care models across a network of facilities?	To what extent is the hospital actively involved in local, regional or national networks aimed at developing integrated care models across a network of facilities?	1 = Not at all 2 3 4 5 6 7 = To a full extent	[10,23,24,25,37]
Change management section Implementing VBHC principles within hospital dynamics require structural changes and, more importantly, cultural transformations. Indeed, for the principles of this theory to become more than just another innovation proposed within the healthcare sector and truly take root within hospital dynamics, it is essential that healthcare professionals understand and support them, and become the leading actors in this transition process. Therefore, this section aims to answer the following questions: what change management strategies did top management employ to facilitate the transition to the VBHC model?					
Establishing a sense of urgency	14	To what extent did your organization establish a sense of urgency to implement the principles of VBHC?	To what extent did your organization establish a sense of urgency to implement the principles of VBHC?	1 = Not at all 2 3 4 5 6 7 = To a full extent	[16,23,43,44,46,48]
	15	If so, which of the following factors push up the urgency level? (Select all that apply)	If so, which of the following factors push up the urgency level? (Select all that apply)	Political or regulatory pressures Drive from hospital top management Unsatisfactory clinical outcome data Patients feedback Initiative of healthcare professionals Financial challenges Market pressure I do not know Other factors (please specify)	[16,23,43,45,46,48]
Creating a guiding coalition	16	To what extent was a formal group of people—representatives from clinical and managerial teams—brought together to drive the implementation of VBHC?	To what extent was a formal group of people—representatives from clinical and managerial teams—brought together to drive the implementation of VBHC?	1 = Not at all 2 3 4 5 6 7 = To a full extent	[14,16,25,27,28,32,33,37,38,39,40,41,49,50]
Developing and communicating a vision and strategy	17	To what extent has an official vision for VBHC (“a future-oriented perspective that emphasizes the reasons why you should strive to create that future”) been established within the hospital?	To what extent has an official vision for VBHC (“a future-oriented perspective that emphasizes the reasons why you should strive to create that future”) been established within the hospital?	1 = Not at all 2 3 4 5 6 7 = To a full extent	[16,27,29,32,40,41]
	18	If so, which of the following activities have been conducted to promote and disseminate the VBHC vision within the hospital? (Select all that apply)	If so, which of the following activities have been conducted to promote and disseminate the VBHC vision within the hospital? (Select all that apply)	Department meetings Workshops Written documentation (e.g., email, brochures, reports) Internal communication channels (e.g., intranet, bulletin boards) Other (please specify)	[16,27,29,32,40,41]
	19	To what extent has a tailored business plan (“a detailed document that outlines your hospital’s goals on VBHC, the strategy for achieving them, and the resources required”) been developed to provide a clear, structured, goal-oriented process?	To what extent has a tailored business plan (“a detailed document that outlines your hospital’s goals on VBHC, the strategy for achieving them, and the resources required”) been developed to provide a clear, structured, and goal-oriented process?	1 = Not at all 2 3 4 5 6 7 = To a full extent	[16,24,28,33,40,41]
	20	To what extent are members of the organization regularly updated on the progress of the VBHC plan over time?	To what extent are you informed about the progress of such VBHC plan over time?	1 = Not at all 2 3 4 5 6 7 = To a full extent	[14,16,26,31,32,37,40,50]
Anchoring new approaches in the culture	21	To what extent have educational initiatives been organized for employees of the organization on the VBHC principles?	To what extent have educational initiatives been organized for you on the VBHC principles?	1 = Not at all 2 3 4 5 6 7 = To a full extent	[14,16,24,27,31,32,33,34,39,40]
	22	If so, which of the following educational initiatives have been used? (Select all that apply)	If so, which of the following educational initiatives have been used? (Select all that apply)	Training courses Workshop and seminars Informational materials (e.g., PowerPoint presentations) Visits of early adopters Other activities (please specify):	[14,16,24,27,31,32,33,39,40]
	23	To what extent has the organization involved the key internal stakeholders (e.g., employees, management teams, clinical staff) in the change process?	To what extent do you feel involved in the change process?	1 = Not at all 2 3 4 5 6 7 = To a full extent	[16,24,26,31,32,35,40,50]
	24	To what extent do employees have the time to work on VBHC activities and anchor changes into their daily work?	To what extent do you have the time to work on VBHC activities and anchor changes into your daily work?	1 = Not at all 2 3 4 5 6 7 = To a full extent	[16,28,32,33,37,38]
Commitment to the change program	25	To what extent do you believe that the changes introduced by the VBHC model will be your way of delivering care in the long term?	To what extent do you believe that the changes introduced by the VBHC model will be your way of delivering care in the long term?	1 = Not at all 2 3 4 5 6 7 = To a full extent	[16,29,32,38]
	26	Which of the following strategies was utilized to support the commitment toward VBHC of the employees within the hospital? (Select all that apply)	Which of the following strategies have been utilized to support your commitment toward VBHC? (Select all that apply)	Career progression opportunities Increased visibility within the hospital Financial incentives Educational incentives Strengthening of staff (e.g., hiring additional resources or providing additional support) Other activities (please specify) No strategy was employed	[16,17,51]
	27	How would you describe the current level of commitment of the employees of the organization?	How would you describe your current level of commitment?	Active participation in the change process, with a strong commitment to adopting VBHC Support for the change, even without direct involvement Neutrality towards the change with minimal involvement Participation driven more by obligation than genuine engagement Perception that the change stems from top-down decisions, making individual input seem irrelevant Concerns about how the change is being managed and Other (please specify)	[8,14,16,24,25,27,28,32,33,35,37,38,39,40,41,49,50,51,52]
	28	To what extent have initial successes or challenges/failures, during the change process toward VBHC, impact the level of commitment among the members of the organization?	To what extent have initial successes or challenges/failures, during the change process toward VBHC, impact your level of commitment?	1 = Not at all 2 3 4 5 6 7 = To a full extent	[16,29,32,38]
	29	Which strategies have been implemented to maintain engagement and motivation of the employees of the organization after facing challenges, failures, or initial successes in implementing VBHC? (Select all that apply)	Which strategies have helped maintain your engagement and motivation after facing challenges, failures, or initial successes in implementing VBHC? (Select all that apply)	Analysis and discussion of failures Clear communication of progress Reinforcement of the vision and goals of the change Additional training Supportive leadership and management presence Team-building activities or events Providing incentives or rewards Adjustments to workload or responsibilities to ease transition Other strategies (please specify)	[16,29,32,38]
Removing obstacles	30	Which were the main obstacles you encountered during the implementation of VBHC? (Select all that apply)	Which were the main obstacles you encountered during the implementation of VBHC? (Select all that apply)	Resistance to change among staff Lack of alignment or support from top management Limited financial resources or budget constraints. Unclear communication process Lack of incentive systems Lack of IT support Others	[16]

## Data Availability

The original contributions presented in this study are included in the article/Appendix A. Further inquiries can be directed to the corresponding author.

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
