# Peer review of "Development of a Tool to Assess the Implementation and Perception of the Value-Based Healthcare Model: Our Results"

_healthcare, 2025, doi:10.3390/healthcare13222849_

Round 1

Reviewer 1 Report

Comments and Suggestions for Authors

Dear authors,

You have conducted a study on the implementation of a multi-professional tool to assess perceptions of the Value-Based Healthcare model. To this end, you have developed a qualitative methodology that is well described and based on a previous study.

Your manuscript is interesting but I need you to answer some questions:

ABSTRACT

  • The authors have written extensively about the introduction and very little about the methodology.
  • The conclusions are also very long and the results brief.

INTRODUCTION

  • The objective is not well formulated. It should clearly state the purpose, not refer to something mentioned previously.

MATERIALS AND METHODS

  • Was there any analysis of the responses between round 1 and round 2? The authors should explain the procedure in greater detail.

RESULTS

3.2. Tool validation.

  • The authors previously discussed the estimated response rate. They must now specify the actual response rate.
  • There is redundant information in “Materials and methods”.

REFERENCES

  • Many bibliographies are obsolete. The bibliographic citations used are more than 5 years old (45.3 %). The authors must update and arrange the bibliography.
  • Some references are incomplete or have errors. The authors should review this section.

Author Response

Dear reviewer,

I would like to express my gratitude for your suggestions. I have taken your suggestions and incorporated them into the text, highlighting them in red.

ABSTRACT

  • The authors have written extensively about the introduction and very little about the methodology
  • The conclusions are also very long and the results brief 

Response: Dear reviewer, thank you for the suggestions. This error has been fixed

INTRODUCTION

  • The objective is not well formulated. It should clearly state the purpose, not refer to something mentioned previously.

Response: Dear reviewer, thank you for the suggestions. We have added it in lines 89-94.

MATERIALS AND METHODS

  • Was there any analysis of the responses between round 1 and round 2? The authors should explain the procedure in greater detail.

 Response: Dear reviewer, thank you for the suggestions. We have added it in lines 143-146.

RESULTS

  • The authors previously discussed the estimated response rate. They must now specify the actual response rate.

Response: Dear reviewer, thank you for the suggestions. We have added it in line 165.

  • There is redundant information in “Materials and methods”.

Response: Dear reviewer, thank you for the suggestions. I believe we have removed the various redundant information elements in "Materials and methods."

REFERENCES

  • Many bibliographies are obsolete. The bibliographic citations used are more than 5 years old (45.3 %). The authors must update and arrange the bibliography.
  • Some references are incomplete or have errors. The authors should review this section.

Response: Dear reviewer, thank you for the suggestions. This error has been fixed

Reviewer 2 Report

Comments and Suggestions for Authors

Dear authors,

First of all, congratulations on choosing the topic of the development and validation of a multi-professional tool for assessing the implementation and perception of a value-based healthcare (VBHC) model. You have put in a lot of effort, which is very visible in the article, but before publishing, I would like to suggest a few improvements that will, I hope, additionally make your work easier to understand for readers. First, I would like to ask you to harmonize the key figures and thresholds from Delphi throughout the text. In the abstract and methods, you state two different numbers of invited experts, I recommend that the same number (for example, 54) be used consistently and that it be clearly described how many of them participated in each round. Also, in the Methods, you define the consensus criteria (median, IQR and percentage threshold), while in the Abstract you claim that all items reached >80% in both rounds. If this is indeed the case, it is worth highlighting it, if not, it is better to present the actual percentages by units and maintain a consistent formulation. The following is a set of recommendations of a methodological and statistical nature. Please specify which “K-coefficient” was used (Kendall’s W, Cohen’s κ or something else), and show the values ​​for it by rounds and, if possible, by instrument domains. Where you talk about testing differences between rounds, it is advisable to consistently use the term “Mann Whitney U test for independent samples” and provide p-values ​​for key comparisons, with a brief explanation of what this means for the stability of the scores. You mention RIR/VRIR, coefficient of variation and related stability indicators, but the formulas and sources are not provided, my suggestion is to briefly define them and refer the reader to the appropriate references, and move the extended tables to the Appendix. I also note that Cronbach’s α is very high, it would be useful to say whether you considered removing redundant items after analyzing “item total” correlations and whether this increase in α was negligible (which would confirm the retention of all items). Also, I recommend that you write a clearer description of the criteria for selecting experts, the method of invitation and reminders, and a brief comment on geographical representation. Since most of the participants are from one country, it is good to state this fact in the Limitations and suggest that future validations include a broader international base of respondents and a higher proportion of non-clinical stakeholders such as managers, health economists, patient organizations. It is impressive that you are already clear about the scope of the tool, it does not try to cover all aspects of VBHC, but rather focuses on the implementation and perception of key components. I suggest that you also transfer this “scope statement” to the summary in one short line, especially since you consciously omit systemic elements such as payment-for-value and platform-based learning. I wish you all the best in your future work and publication of this article.

Kind regards

Author Response

Dear reviewer,

I would like to express my gratitude for your suggestions. I have taken your suggestions and incorporated them into the text, highlighting them in red.

  • First, I would like to ask you to harmonize the key figures and thresholds from Delphi throughout the text. In the abstract and methods, you state two different numbers of invited experts, I recommend that the same number (for example, 54) be used consistently and that it be clearly described how many of them participated in each round.

Response: Dear reviewer, thank you for the suggestions. This error has been fixed.

  • Also, in the Methods, you define the consensus criteria (median, IQR and percentage threshold), while in the Abstract you claim that all items reached >80% in both rounds. If this is indeed the case, it is worth highlighting it, if not, it is better to present the actual percentages by units and maintain a consistent formulation.

Response: We have aligned the Abstract and Methods. The Abstract now explicitly states the consensus rule and reports that all items achieved ≥80% agreement in both rounds. Item- and domain-level percentages are reported consistently in the Results and in Table S1 (Appendix A).

  • Please specify which “K-coefficient” was used (Kendall’s W, Cohen’s κ or something else), and show the values for it by rounds and, if possible, by instrument domains.

Response: Thank you for the clarification request. We now explicitly state that the K-coefficient refers to Kendall’s coefficient of concordance (W). Values are reported by round and by domain in the Results and in Table S1 (Appendix A), with a brief interpretive commentar

  • Where you talk about testing differences between rounds, it is advisable to consistently use the term “Mann Whitney U test for independent samples” and provide p-values for key comparisons, with a brief explanation of what this means for the stability of the scores.

Response: We consistently use the term “Mann–Whitney U test for independent samples” to compare rounds and report p-values for key item/domain comparisons. Results are summarized in the text and provided in Table S1 (Appendix A), with a brief note on what these findings imply for score stability.

  • You mention RIR/VRIR, coefficient of variation and related stability indicators, but the formulas and sources are not provided, my suggestion is to briefly define them and refer the reader to the appropriate references, and move the extended tables to the Appendix.

Response: We revised Section 2 (“Consensus evaluation between rounds”) to briefly define each stability indicator and provide the corresponding formulas (IQR, RIR, VRIR, CV, VCV). As methodological support, we cite Mengual-Andrés, Roig-Vila, & Mira (2016) (already in the reference list).

  • I also note that Cronbach’s α is very high, it would be useful to say whether you considered removing redundant items after analyzing “item total” correlations and whether this increase in α was negligible (which would confirm the retention of all items).

Response: We acknowledge that Cronbach’s α is very high (0.962), which may suggest redundancy. We examined corrected item–total correlations (all ≥ 0.30) and “alpha if item deleted” for all items; no deletion produced more than trivial changes in α. Given the scale’s conceptual coverage and negligible gains from removal, all items were retained, as now clarified in the manuscript.

  • I recommend that you write a clearer description of the criteria for selecting experts, the method of invitation and reminders, and a brief comment on geographical representation.

Response: Dear reviewer, thank you for the suggestions. We have added it in lines 123-133.

  • Since most of the participants are from one country, it is good to state this fact in the Limitations and suggest that future validations include a broader international base of respondents and a higher proportion of non-clinical stakeholders such as managers, health economists, patient organizations

Response: Dear reviewer, thank you for the suggestions. We have added it in lines 356-361.

  • It is impressive that you are already clear about the scope of the tool, it does not try to cover all aspects of VBHC, but rather focuses on the implementation and perception of key components. I suggest that you also transfer this “scope statement” to the summary in one short line, especially since you consciously omit systemic elements such as payment-for-value and platform-based learning

Response: Dear reviewer, thank you for the suggestions. We have added it in lines 320-323 and 347-361.

Reviewer 3 Report

Comments and Suggestions for Authors

Thank you for the opportunity to review the manuscript. Please, find my comments.

The article is well organized.

The objectives of the study are clearly presented.

The results and discussion are consistent.

However, the tittle could be more attractive.

I suggest describing the acronym “CVs” (Line155).

The initial set of questions could be presented as supplementary material.

Author Response

  • I suggest describing the acronym “CVs” (Line155)

Response: Dear reviewer, thank you for the suggestions. This error has been fixed.

  • The initial set of questions could be presented as supplementary material

Response: Dear reviewer, thank you for the suggestions. Items that met the inclusion criteria in the first round were retained unchanged and re-evaluated in the second round to verify the stability of expert consensus. No modifications were made between rounds;

Reviewer 4 Report

Comments and Suggestions for Authors

STRENGTHS

  • The manuscript addresses a highly relevant topic: the operationalization of Value-Based Healthcare (VBHC) through a multi-professional assessment tool.
  • The tool development process is robust, clearly grounded in a prior scoping review and structured through a two-round Delphi method.
  • The mirror-survey design (managers vs. clinicians) is an innovative feature, offering a comprehensive perspective on organizational alignment.
  • The statistical analyses used (IQR, Cronbach’s alpha, Mann–Whitney U) are appropriate and well-explained, reinforcing the reliability and stability of the tool.
  • The discussion effectively positions the tool within the existing literature, highlighting differences with prior instruments.

LIMITATIONS

  • There is no external validation or pilot implementation in real hospital settings to assess feasibility or predictive validity.
  • “Limitations” and “Practice Implications,” could be more concise and better connected to the study’s main objectives.

RECOMMENDATIONS

  • Consider expanding the Delphi panel in future iterations to include a more balanced international sample and possibly patient representatives.
  • In the discussion, expand on the implications of the missing pillars, clarifying how future versions of the tool might integrate them or why their exclusion is justified.
  • Add a graphical summary or framework figure of the tool (two macro-areas + mirror design) to help readers visualize its structure quickly.
Comments on the Quality of English Language
  • Language is occasionally dense, especially in the Introduction, and could benefit from stylistic refinement to improve readability (long sentences, repetition of key concepts....)
  • Revise the manuscript for linguistic clarity and conciseness, especially in the introduction and discussion, to make it more accessible to a broader readership.

Author Response

  • Consider expanding the Delphi panel in future iterations to include a more balanced international sample and possibly patient representatives

Response: Dear reviewer, thank you for the suggestions. We have added it in lines 347-361.

  • In the discussion, expand on the implications of the missing pillars, clarifying how future versions of the tool might integrate them or why their exclusion is justified.

Response: Dear reviewer, thank you for the suggestions. We have added it in lines 320-323 and 347-361.

  • Add a graphical summary or framework figure of the tool (two macro-areas + mirror design) to help readers visualize its structure quickly

Response: Dear reviewer, thank you for the suggestions. We have added it in line 286.

  • Comments on the Quality of English Language

Language is occasionally dense, especially in the Introduction, and could benefit from stylistic refinement to improve readability (long sentences, repetition of key concepts....)

Revise the manuscript for linguistic clarity and conciseness, especially in the introduction and discussion, to make it more accessible to a broader readership

Response: Dear reviewer, I would like to express my sincere gratitude for your feedback. We believe we have solved these mistakes.

Round 2

Reviewer 1 Report

Comments and Suggestions for Authors

Dear authors,

Thanks for your reply. The explanations that you provide are satisfactory. The paper has greatly improved its quality.

Congratulations on your work.

Best regards

Author Response

Dear Reviewer, thank you very much for your valuable feedback.

Reviewer 2 Report

Comments and Suggestions for Authors

Dear Authors,

thank you for considering my suggestions to improve the quality of this manuscript.

Good luck with the publication.

Author Response

(The authors gave the same response as above.)

Reviewer 3 Report

Comments and Suggestions for Authors

The article is well organized. Please, find my suggestions.

-Title: “A multi-professional perspective tool to assess the implementation and perception of the Value-Based Healthcare (VBHC) model: tool development and Delphi results”.

The tittle could be more attractive and simple. (For example: “Development of a tool to assess the implementation and perception of the Value-Based Healthcare Model: our results”)

-Line 56:”… integrated IT platforms to coordinate all components…”. Please, describe the acronym “IT”. 

Lines 123-124:  “Potential members of the Delphi panel were selected among international experts based on their Curriculum Vitae, scientific publications, and demonstrated expertise,,,” OR  “For the Delphi panel, potential members, who spoke English, were selected among international experts based on their Curriculum Vitae, scientific publications, and demonstrated expertise,,,”  Please, think about this.

-Lines 178-179: “A total of 54 experts were invited to participate in the Delphi survey. The first round received 25 responses (46%), while the second round had 23 responses (42%) (Table 1). However, the table 1 only describes the panel composition and expertise of the first round. Please, describe the panel composition of the second round.

-Line 179: “...25 responses (46%), while the second round had 23 responses (42%) ...”  OR  “25 responses (46%), while the second round had 23 responses (43%) …)? ” Please, correct the information.

-Lines 178-179: “A total of 54 experts were invited to participate in the Delphi survey. The first round received 25 responses…” /Line 193: “The level of agreement among the 13 experts, measured using Kendall’s…” Why did the authors write 13 experts and not 25? Please, explain it in the text.

-Line 179: “…while the second round had 23 responses…” / Line 211: “…among the 17 experts, assessed using Kendall’s coefficient of concordance…” …” Why did the authors write 17 experts and not 23? Please, explain it in the text.

-The authors could move Figure 3 to “Results” section.

-The text in Figure 3 has letters that are too small to read.

-Line 291: “…Each area includes "mother questions"…”   OR “Each area includes "main questions"…” Please, think about this.

-Line 292: “…by “child questions"…”  OR  “…by “secundary questions"…”  Please, think about this.

Author Response

Dear reviewer,

I would like to express my gratitude for your suggestions. I have taken your suggestions and incorporated them into the text, highlighting them in red.

-Title: “A multi-professional perspective tool to assess the implementation and perception of the Value-Based Healthcare (VBHC) model: tool development and Delphi results”.

The tittle could be more attractive and simple. (For example: “Development of a tool to assess the implementation and perception of the Value-Based Healthcare Model: our results”)

 Response: Dear reviewer, thank you for the suggestions. We changed the title of the paper based on your suggestion.

-Line 56:”… integrated IT platforms to coordinate all components…”. Please, describe the acronym “IT”. 

Response: Dear reviewer, thank you for the suggestions. This error has been fixed.

Lines 123-124:  “Potential members of the Delphi panel were selected among international experts based on their Curriculum Vitae, scientific publications, and demonstrated expertise,,,” OR  “For the Delphi panel, potential members, who spoke English, were selected among international experts based on their Curriculum Vitae, scientific publications, and demonstrated expertise,,,”  Please, think about this.

 Response: Dear reviewer, thank you for the suggestions. This error has been fixed.

-Lines 178-179: “A total of 54 experts were invited to participate in the Delphi survey. The first round received 25 responses (46%), while the second round had 23 responses (42%) (Table 1). However, the table 1 only describes the panel composition and expertise of the first round. Please, describe the panel composition of the second round.

 Response: Dear reviewer, thank you for the suggestions. This error has been fixed.

-Line 179: “...25 responses (46%), while the second round had 23 responses (42%) ...”  OR  “25 responses (46%), while the second round had 23 responses (43%) …)? ” Please, correct the information.

 Response: Dear reviewer, thank you for the suggestions. This error has been fixed.

-Lines 178-179: “A total of 54 experts were invited to participate in the Delphi survey. The first round received 25 responses…” /Line 193: “The level of agreement among the 13 experts, measured using Kendall’s…” Why did the authors write 13 experts and not 25? Please, explain it in the text.

 Response: Dear reviewer, thank you for the suggestions. There was a typo. 13 were not the experts but the items.

-Line 179: “…while the second round had 23 responses…” / Line 211: “…among the 17 experts, assessed using Kendall’s coefficient of concordance…” …” Why did the authors write 17 experts and not 23? Please, explain it in the text.

 Response: Dear reviewer, thank you for the suggestions. There was a typo. 17 were not the experts but the items.

-The authors could move Figure 3 to “Results” section.

 Response: Dear reviewer, thank you for the suggestions. This error has been fixed.

-The text in Figure 3 has letters that are too small to read.

  Response: Dear reviewer, thank you for the suggestions. This error has been fixed.

-Line 291: “…Each area includes "mother questions"…”   OR “Each area includes "main questions"…” Please, think about this.

  Response: Dear reviewer, thank you for the suggestions. This error has been fixed.

-Line 292: “…by “child questions"…”  OR  “…by “secundary questions"…”  Please, think about this.

 Response: Dear reviewer, thank you for the suggestions. This error has been fixed.

Reviewer 4 Report

Comments and Suggestions for Authors

The manuscript has been sufficiently improved.

Comments on the Quality of English Language

The manuscript has been sufficiently improved to warrant publication.

Author Response

(The authors gave the same response as above.)
